# Recent Developments in Oral Delivery of Vaccines Using Nanocarriers

**DOI:** 10.3390/vaccines11020490

**Published:** 2023-02-20

**Authors:** Amna Zafar, Raffia Arshad, Asim Ur.Rehman, Naveed Ahmed, Hashaam Akhtar

**Affiliations:** 1Department of Pharmacy, Quaid-i-Azam University, Islamabad 45320, Pakistan; 2Yusra Institute of Pharmaceutical Sciences, Yusra Medical and Dental College, Islamabad 45730, Pakistan

**Keywords:** vaccine, nanoparticles, oral delivery, polymeric NPs, liposomes, advanced vesicular systems, immune response

## Abstract

As oral administration of vaccines is the preferred route due to its high patient compliance and ability to stimulate both cellular and humoral immune responses, it is also associated with several challenges that include denaturation of vaccine components in the acidic environment of the stomach, degradation from proteolytic enzymes, and poor absorption through the intestinal membrane. To achieve effective delivery of such biomolecules, there is a need to investigate novel strategies of formulation development that can overcome the barriers associated with conventional vaccine delivery systems. Nanoparticles are advanced drug delivery carriers that provide target-oriented delivery by encapsulating vaccine components within them, thus making them stable against unfavorable conditions. This review provides a detailed overview of the different types of nanocarriers and various approaches that can enhance oral vaccine delivery.

## 1. Introduction

The year 2020 was a devastating one for the healthcare systems of the world. A previously unknown variant of a known virus spread rapidly worldwide and quickly became one of its leading killers [1,2]. The biggest medical advancement in recent years has been the development of vaccines that have saved more lives than any other treatment option [3]. In terms of scope, scale, and speed, the worldwide effort to develop vaccines in response to the COVID-19 pandemic is unmatched, and almost no stone was left unturned during the research and development (R&D) phases [4]. Vaccination is a constructive combo pack option for the community, as it works for the immunization of the vaccinated population, as well as generating herd immunity for the unvaccinated proportion [5]. Although the basics of vaccinology lie with the concept of Edward Jenner’s smallpox vaccine development back in 1796, the latest advancements have made them safer and more diversified. Scientists have been able to develop multi-targeted Penta variants or even anticancer vaccines [6,7].

If we focus on the traditional definition of vaccines only, they are always on a mission to either boost immunity against pathogens or to cut off the paths initiated by them. They are biological preparations that can provide active acquired immunity against particular infectious diseases through antigen recognition procedures [6]. The procedure for either the development or the delivery of a vaccine has always been complicated, which has involved a lot of money and time to reach the market after passing all safety issues [8]. It is a known fact that vaccines involve immunization with the whole (live or killed) microorganisms or microbial components of the pathogen, which may have the worst adverse effects one can either think of while delivering it parenterally or orally (PO) [9].

Low patient compliance related to invasive, inconvenient, or unpleasant routes using injections for vaccination has been a major factor in vaccine hesitancy for a long time. Hence, other routes for vaccine administration to raise patient compliance in both primary immunizations and subsequent boosts are being increasingly investigated [3,10]. The oral route is one of the first routes for administering therapeutics, but due to the unpleasant environmental state of the gut, it has been proven to be a difficult one when it comes to biopharmaceutical products such as insulin, vaccines, or other chemokines [9]. Still, the oral delivery of therapeutics represents the current gold standard due to the opportunity for self-administration, safety, improved patient compliance, and ease of distribution compared to injection-based therapies [10,11].

Nevertheless, there are several approaches that have been explored to deliver biopharmaceutical agents through the oral route. As discussed earlier, the oral route is not amenable to the delivery of many proteins and peptides, and advancements in drug delivery systems have changed the situation for vaccine delivery in many cases, including mRNA vaccine technologies [12]. The selection of the route follows the selection of the drug disposal site and, in the case of oral delivery, the gut lymphoid tissue, which is found throughout the mammalian intestine and has membranous (M) cells on its luminal surface that provide a portal of entry for antigens, making it a suitable site of action [13].

### 1.1. Oral Delivery of Vaccines

The most preferred and well-liked mode of administration is oral delivery. Despite this, due to the inherent challenges posed by the digestive system, only a tiny portion of presently approved vaccines are oral formulations. Each portion of the GI tract offers a challenge with a different form of engineering that needs to be carefully handled to create an effective vaccination design [14]. Most pathogens (~90%) enter the body through the mucosal pathway, which includes the respiratory, GI, and urogenital systems, The body’s mucosal systems are more susceptible to infections than the skin due to their huge, thinner, and exposed surface area and more permeable mucosal membrane barriers. An efficient mucosal immune response is thus preferred as a first line of defense to completely defend against infections and the effects of their toxins on the mucosal surface. A significant immunoglobulin called secretory IgA (SIgA) is essential and present for preventing pathogen invasion at mucosal sites [15]. Table 1 shows a list of oral vaccines that are available in the market and those that are under trial.

### 1.2. Yeast-Based Vaccines

Advances in protein chemistry and biology permit the expression of peptides and proteins in eukaryotic cells of mammals, plants, animals, and fungi, as well as prokaryotic bacteria. Among all, yeast/single-celled fungi represent a model of choice for the expression of medically important proteins.

Among yeast and single-celled fungi, *Saccharomyces cerevisiae*, *Kluyveromyces lactis*, and *Pichia pastoris* have the highest efficacy of heterologous gene expression.

Despite being nonpathogenic, yeast cells used as vectors of vaccines can elicit an immune response in mammals and can be taken up by macrophages and dendritic cells [18].

### 1.3. Adenoviral-Based Vaccines

As we discussed yeast as a vector for vaccination, another important vector that can be used for vaccines is adenovirus. Adenoviruses, in general, do not cause considerable damage to human health. They have been used as a study model for eukaryotic cells, e.g., RNA splicing. Their molecular biology has been extensively studied, which has made them an excellent choice to become gene vector antigens. Since they can induce T-cell and B-cell immunologic responses to transgenes, they are suitable candidates for vaccine vectors. Their safety as vaccine carriers is established, as the risk of reversion to virulence is very slim in attenuated pathogens [19].

## 2. Pros and Cons of Oral Delivery of Vaccines

Numerous academic and biopharmaceutical researchers are working to create an alternative, needle-free oral vaccination technology in response to the drawbacks of injectable vaccines [3]. Oral vaccine delivery has several advantages, such as:Improved patient compliance:

Oral vaccine delivery is often simpler and less unpleasant, which may increase the probability that patients would request vaccination and return for follow-up shots [10].

Capacity for mass immunization [20]

Self-administration: Due to their simplicity of administration and ability for self-administration, oral vaccinations have the potential to increase distribution compared to conventional injections. Self-administration reduces the requirement for qualified healthcare staff, making it perfect for extensive and quick vaccinations. Since the cost of launching a new vaccine can account for up to 25% of the cost, this might further cut the cost of vaccination programs [14].

Simplified production and storage [20]

Low production cost: The gut, which is already populated with commensal bacteria, enzymes, nutrients, and lumens, will be exposed to oral vaccination. Therefore, unlike their injectable counterparts, oral vaccines do not need a high level of purity from a scaled-up and manufacturing viewpoint. However, extremely pure vaccines that are endotoxin-free must be used for injectable vaccines [20].

No needle-associated risks: Each year, 5% of healthcare professionals experience needle-related danger, putting them at risk for blood-borne infectious diseases, including HIV/AIDS and hepatitis [14].Both IgG- and IgA-specific response: Antigen-specific mucosal secretory IgA antibodies and antigen-specific systemic IgG antibodies can be produced by vaccinations given by mucosal routes at all mucosal locations, not only the site of delivery [21].

Nevertheless, despite being the most popular delivery method, oral administration has many challenges when reaching immune cells [3]. There are a number of reasons why a generic technology for oral vaccination is lacking [10].

An oral vaccination must first be exposed to a very acidic pH, proteolytic enzymes, and bile salts, which will cause it to degrade in the digestive system (GIT) [22].For successful absorption and penetration across the intestinal walls, vaccinations must pass a variety of biological obstacles (such as the existence of tight epithelial cellular junctions and a thick mucous layer) in the intestinal lumen [22].Vaccines taken orally need to be absorbed and delivered to the right immune system cells. Typically, this is inefficient and requires the administration of high doses of vaccines [23,24].Additionally, the brief antigenic exposure period to mucosal tissues contributes to the decreased absorption of antigenic particles. As a result, compared to their systemic equivalents, oral vaccinations may need multiple and larger doses to have a powerful and long-lasting immunogenic impact [22].However, oral administration of larger and repetitive oral doses of antigens might induce systemic nonresponsiveness (oral tolerance), in which the antigens can be recognized as food or normal flora instead of eliciting protective immune responses [25,26].Another significant obstacle to the development of oral vaccinations is the scarcity of powerful immunostimulants or mucosal adjuvants with minimal toxicity [27].A significant barrier to oral vaccination is the difficulty in measuring the real intensity of the immune response, particularly IgA, at different mucosal sites following oral administration [21].A short vaccination half-life might result from enzymatic breakdown, among other issues. Therefore, the scientific challenge is to significantly increase oral vaccination absorption from the conventional 1% [28].

## 3. Approaches to Enhance Oral Delivery of Vaccines

Different options for the oral delivery of vaccines are present nowadays, as shown in Figure 1, but to enable passive targeting of desired cells, antigen delivery systems’ physical and chemical properties, viz., their size, shape, surface charge, and hydrophobicity, need to be adjusted [29,30]. A number of ligands, such as bacterially produced moieties, lectins, pathogen-associated molecular pattern molecules (PAMPs), and antibodies, have been investigated for targeting vaccine delivery through receptors on intestinal epithelial cells, M cells, and APCs [31,32].

### 3.1. Oral Adjuvants

Commensal bacteria and food antigens are both relatively abundant in the gut environment. Intestinal epithelial cells (IECs) that were formerly thought to just serve as a physical barrier to the outside world are now known to play a more important role in the induction of innate and adaptive immunity and in the maintenance of immunological homeostasis. IECs are the pioneer cells encountering intestinal pathogens, and they can work as immunological sensors detecting pathogens through different classes of pattern recognition receptors (PRRs), including nucleotide-binding oligomerization domain (NOD)-like receptors (NLRs) and Toll-like receptors (TLRs) [32].

Ligands such as wheat germ agglutinin (WGA) that target N-acetyl-D glucosamine and sialic acid residues expressed by enterocytes throughout the GI tract have been well explored for oral drug delivery [14]. The glycolipid galactosylceramide (-GalCer) is an effective inducer of natural killer T cells (NKT). It acts as an adjuvant to elicit T-cell immune response to viruses [33].

A study conducted by Davitt et al. concluded that α-GalCer strongly enhanced antigen-specific antibody responses following oral coadministration with Dukoral^®^ in a mice model [34].

### 3.2. Targeting M Cells of Intestinal Epithelium

M cells might be effective delivery vehicles for oral vaccines [35]. The follicle-associated epithelium (FAE) that rests on top of the organized mucosal-associated lymphoid tissue (O-MALT) found along the whole length of the digestive system contains intestinal M cells [36,37]. Contrarily, a diverse range of diseases use M cells as a pathway for host invasion. It has been proposed that targeting intestinal M cells particularly with vaccine delivery mechanisms may improve the effectiveness of oral vaccines [38,39]. This might be accomplished by covering delivery vehicles with ligands that attach to M-cell surfaces with specificity. Since M cells may be distinguished from enterocytes by the production of cell surface carbohydrates, lectins are one kind of ligand that may allow M-cell targeting of oral vaccines [40,41]. It should also be emphasized that certain lectins are poisonous and can have innate immunogenic properties. Although the immunostimulatory potential of lectins as mucosal adjuvants may be favorable, it also carries the danger of inducing a reaction against the targeted molecule and eventually inhibiting absorption [42].

## 4. Nanoparticle-Based Oral Vaccination Strategies

Nanotechnology holds great promise in the oral delivery of vaccines by incorporating them within the nanoparticulate carrier, which not only improves their stability against proteases but also helps in rapid absorption through the gastrointestinal tract (GIT) [43]. An ideal carrier system should sufficiently protect vaccines/antigens from the unfavorable environment of the GIT and transport antigens to the intended site to protect the body from pathogens [44]. To date, most oral vaccines are delivered as live-attenuated/inactivated microorganisms combined with adjuvants, and they might have the possibility to reverse to an active state, especially in patients having weak immunity conclusively harming the host. Nanoparticles (NPs) have the ability to stimulate a stronger immune response than conventional adjuvants such as alum without the addition of antigen-presenting cells targeting ligands to stimulate immunogens [3]. Vaccines can be enclosed or attached to the surfaces of nanoparticles for adequate exposure to antigen-presenting cells. Additionally, NPs have various peculiar manageable features such as their particle size, surface charge, surface area, high loading capacity, protection in gastric fluid, and enhanced permeation capacity across the mucosal barrier of the intestine [45]. To enhance targeting to immune cells and to change the antigen release behavior from the carrier, active moieties can be attached to the nanoparticulate carrier [46].

The current review will provide a description of the different nanocarriers that can be used for oral vaccine delivery, as shown in Figure 2.

i.Polymeric NPs;ii.Lipid-based systems;iii.Inorganic NPs;iv.Niosomes;v.Advanced vesicular drug delivery systems.

**Figure 2 vaccines-11-00490-f002:**
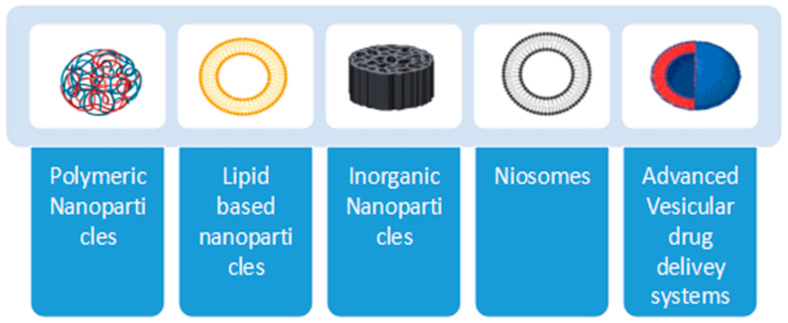
Different types of nanocarriers.

Transferosomes and ethosomes also come under the class of advanced vesicular drug delivery systems, but these systems are not discussed in detail in this review, as transferosomes and ethosomes majorly show applications in transdermal drug delivery and have very limited application in the oral delivery of drugs [47].

### 4.1. Polymeric NPs

These are small-sized particulate carriers that can encapsulate, adsorb or conjugate any antigen into their core or on the outer surface [48]. Over a period of twenty years, several polymers have been investigated for their application in vaccine delivery. As biological molecules are sensitive and can be deteriorated while passing through the GIT, encapsulation of vaccines within polymers provides protection to antigens and control over the release at the site [49]. Surface modification of these polymeric nanoparticles can be performed to enhance active targeting. Figure 3 shows the different routes of transport of nanoparticles, i.e., transcellular and paracellular, through the intestine when given orally and also describes the activation of humoral and cell-mediated immune responses upon vaccination. First, APC recognizes pathogen or vaccine antigens, and after processing, they are presented on the surface of the major histocompatibility complex (MHC-II). Antigens bound to receptor molecules, i.e., MHC-II, will then attach to the T-cell receptors of naïve CD4 T-cell. Upon activation of T-cell receptors, nïive CD4 T cells differentiate into Th1, Th2, Th9, Th17, Th22, T follicular helper (Tfh), and T-regulatory cells (Treg) under the stimulus of cytokines. Cytokines also stimulate B cells to activate and convert them into plasma cells and memory B cells. Plasma cells will then release antibodies into the blood. Cytotoxic CD8 T cells release granzyme, perforin, and INF-, which causes the destruction of infected host cells. Memory B and T cells are responsible for providing future immunity against similar pathogens or antigens [50].

For targeted delivery, a wide range of polymers with the required physical and chemical properties can be used [51]. Advancements in polymer chemistry have provided us with natural and synthetic polymers that can be used for the oral delivery of drugs/antigens. Polymeric nanoparticles are categorized on the basis of polymer characteristics and the mechanism of target-specific antigen delivery [52]. Such classes are:➢Mucoadhesive NPs;➢Stimuli-responsive NPs;➢Specific ligand-bearing NPs.

Synthetic polymers such as polyesters, i.e., poly lactic acid (PLA) and poly lactic-co-glycolic acid (PLGA), have been most commonly used to prepare biodegradable nanoparticle-based vaccines that provide prolonged antigen release with a single-dose administration [53]. On the other hand, nanoparticles formed using these polymers possess definite challenges such as low loading capacity, hydrophobicity, rapid initial release, being expensive, and difficulty in scale-up [51]. Taking oral application into consideration, Carreno et al. developed an oral vaccine by encapsulating *Salmonella typhi* porin proteins into PLGA. This encapsulation showed enhanced efficacy due to the protection of antigens from degradation in the stomach pH after delivery through the oral route. pH-sensitive polymers protect antigens from the harsh gastric environment and release antigens only in the intestine region [54]. Zhu et al. developed pH-sensitive nanoparticles for oral delivery by incorporating a vaccine into PLGA polymers and further coating these NPs with a pH-sensitive polymer (Eudragit) to target drug delivery in the large intestine for the immunoprotection of rectal and vaginal mucosa [55]. Researchers have tried many ways to elevate the stability of antigens in the low pH of the stomach, and one of those methods was to conjugate polyethylene glycol (PEG) on both sides of PLA, which encapsulates hepatitis B surface antigens, and this encapsulation protects the antigens from degradation in gastric fluids and showed enhanced intestinal uptake. This conjugation provided increased immunogenicity even with a single oral dose, eliminating the need for a booster dose [56]. By contrast, this copolymerization reduced the entrapment efficiency of enclosed antigens [3]. Crosslinking of polymers enables the release of antigens in a controlled manner. In another study, crosslinking of poly-dextran aldehyde with water-soluble polymeric NPs prevented rapid leakage of antigens and immune response, i.e., antibody titers induced by crosslinked NPs were higher than non-cross-linked formulations [57].

Another class of synthetic polymers is polyanhydrides. They are also biodegradable but less acidic than polyesters and help to enhance the stability of entrapped molecules. These polymers enhance immunogenicity without the need for additional adjuvants. Mannan-coated methacrylic acid-based copolymer is also an example of this type of synthetic polymer, which provides a pH-dependent release of entrapped antigens along with the activation of activated phagocyte cells (APCs) [50]. The use of mucoadhesive polymers for the oral delivery of NPs is a novel approach for sustained and targeted drug delivery. Mucoadhesive polymers lengthen the retention time due to adhesive forces and allow extensive exposure of antigens to the intestinal cells for intensified absorption [58]. One study revealed that coating of nanospheres with poly (butadiene-maleic anhydride-co-L-DOPA) enhances nanoparticle uptake 10 times more in the small intestine as compared to uncoated nanospheres, and site-specific delivery can be provided by attaching a specific ligand with a mucoadhesive polymer [50].

A literature survey explained the application of naturally obtained polymers, e.g., chitosan and its derivatives, in the oral delivery of proteins and peptides against ovarian and breast cancer [59]. Chitosan and its derivatives are mucoadhesive in nature [60], and they are capable of provoking an immune response by direct interaction with M cells or by opening the tight junctions within intestinal epithelial cells. The cationic nature of chitosan enables its strong binding with the negatively charged components of mucus, i.e., sialic acid, by electrostatic interactions and prevents the early mucociliary clearance of antigens [61]. As chitosan is only soluble in acidic pH, the weakly acidic to neutral pH of the intestine limits its absorption; hence, by enhancing its solubility in the intestine, the amino groups of chitosan can be methylated, which will enhance its solubility at neutral pH [62]. Oral delivery of antigens can be improved by attaching M-cell-specific targeting ligands to the surface of chitosan nanoparticles. Yoo et al. conjugated chitosan nanoparticles with M-cell-directing peptide as a carrier for oral vaccine delivery. This study showed the target-specific binding of a nanocarrier to M cells, which enhances the intestinal absorption of vaccines [63]. Chitosan NPs can also be used for DNA vaccine delivery and provoke the greater production of interferon-gamma (INF-γ) and interferon-4 (IL-4) as compared to chitosan and IL-2 when given alone. Chitosan NPs can enhance Toll-like receptor (TLR), Th1, and Th2 cytokine mRNA expression in chicken immune cells when Salmonella subunit antigens are administered orally. An avian influenza virus vaccine when enclosed within chitosan NPs can better activate lymphocyte proliferation as compared to when it is loaded within chitosan [64]. Alginate is another example of a natural biodegradable polymer that has an intrinsic property of M-cell targeting and stimulation of immune response [65]. Chitosan alone cannot provide complete protection to entrap antigens due to its high solubility in the gastric pH that causes an immediate burst release of antigens in the stomach. To combat this limitation of chitosan, chitosan nanoparticles can be covered with an acid-resistant polymer, i.e., alginate [66]. Oliveira et al. described the electrostatic interactions between the amino groups and the carboxyl groups of chitosan and alginate, respectively, which results in the formation of a polyelectrolyte complex. Coated chitosan nanoparticles remained stable in simulated gastric fluids. Alginate coating enhanced the entrapment efficiency of chitosan NPs and also prolonged the release of antigens that made them an interesting carrier for the mucosal delivery of vaccines [67].

### 4.2. Lipid-Based Nanoparticles

➢Liposomes

Liposomes have gained immense importance in drug delivery due to their ability to encapsulate both hydrophobic and hydrophilic components. These are composed of a self-assembly of phospholipids constituting an inner aqueous core and an outer lipid bilayer [68]. Hydrophilic antigens can be efficiently entrapped within the core, whereas the hydrophobic components can be incorporated in the lipid bilayer or at the junction of an inner and outer layer. The ability of liposomes to entrap both hydrophilic and lipophilic components makes them an interesting tool for delivering antigens and immunostimulators [69].

Liposomes exhibit many advantages, including particle uptake by M cells and the ability to induce cytotoxic T lymphocytes by targeting antigens to endogenous processing pathways [70]. Modification in the surface properties of liposomes can enhance entrapment efficiency, stability, slow and sustained release, mucoadhesiveness, and immune cell targeting capacity. Besides the positive aspects of liposomes, they also experience some drawbacks that involve low antigen loading capacity and poor stability [71]. Unmodified liposomes become degraded in acidic pH and are easily dissolved by pancreatic lipase. Intestinal bile salts can also impair the integrity of the phospholipid bilayer, resulting in the lysis of liposomes. This lysis causes the early release of entrapped content. The instability of liposomes in GIT can be managed by modifying their formulation. The incorporation of biodegradable and biocompatible bile salts in the composition of liposomes leads to the formation of bilosomes. Bile salts help in the stabilization of liposomes against destruction in the GIT. Aside from increasing the stability of liposomes, sodium glycolate also works as a permeation enhancer and causes an increase in the penetration of liposomes in the intestine. Selective targeting to M cells can be achieved by attaching targeting ligands over the surface of liposomes [72,73].

➢Nanoemulsion

This is an isotropic system formed by adding two immiscible liquids, i.e., oil and water, and is stabilized by using a suitable quantity of surface-active agents to obtain nanosized droplets. Usually, they have a size range of 20 to 200 nm [74]. Size mimicry of nanoemulsions to pathogens enables easy transcytosis across intestinal epithelial cells. Therefore, nanoemulsion-based delivery systems are considered promising carriers for vaccine delivery. Emulsions have the capability to solidify, which facilitates transportation and enhances vaccine stability during storage. Nanoemulsions can be water-in-oil (w/o) and oil-in-water (o/w), and both are capable of delivering various drugs and antigens through the mucosal route. Antigens entrapped within nanoemulsion avoid direct interaction with gastric fluids; hence, they are more stable in the acidic pH of the stomach. Surfactants also play a key role in improving the stability of nanoemulsions from the premature leakage of entrapped antigens in the GIT and also enhance their permeation through membranes [75]. Multiple emulsions have also been studied for the oral delivery of vaccines and are prepared by dispersing w/o or o/w emulsions into the external oily or aqueous phase. These multiple emulsions are then further stabilized by using external surfactants [76]. The w/o/w or o/w/o are the most common types of multiple emulsions. Multiple emulsions are superior to biphasic emulsions, as they can incorporate hydrophilic and hydrophobic antigens in a single formulation and provide better protection to antigens [77]. One study showed the encapsulation of OVA into a squalene oil-based multiple emulsion (w/o/w), and oral administration of this formulation showed better uptake by macrophages and APCs at mucosal lymph nodes, which resulted in a higher immune response by stimulating systemic antibody production. Besides other advantages, emulsions are economical to produce [78].

➢Immunostimulating complexes (ISCOMs)

In 1984, ISCOMs were first studied as carriers for vaccine delivery. They are defined as self-assembled complexes that are spherical in shape. Usually, they have a particle size of 40 nm. Their composition includes antigens, cholesterol, phospholipids, and Quil A saponins [79]. ISCOMs can easily entrap protein-based hydrophobic antigens within the lipid complex, but they are unable to entrap hydrophilic antigens. Chemical modification of hydrophilic antigens is required prior to their inclusion in ISCOMs. They provide strong humoral and cellular immune responses through the major histocompatibility complex I and II pathways against several antigens. ISCOMs are mainly used for parenteral delivery, whereas oral administration of ISCOMs as a vaccine carrier manifests enhanced immunogenicity as compared to soluble antigens. Additionally, ISCOMs avoid tolerance to orally delivered antigens, as they are resistant to bile salts [80].

### 4.3. Inorganic NPs

Inorganic NPs have emerged as novel carriers for the oral delivery of vaccines because of their interesting physicochemical features [81]. Modification in the surface characteristics and particle size (usually 1–100 nm) of inorganic NPs enable enhanced colloidal stability, antigen entrapment efficiency, and target-oriented delivery by attaching specific targeting molecules over them. Inorganic NPs provide protection to entrapped antigens from denaturation in the unsuitable GIT environment and prevent their leakage because of their rigid and stable surfaces [82]. Gold nanoparticles are gaining interest in the delivery of vaccines due to their inert and biocompatible nature [83]. In acidic conditions, gold nanoparticles attached to various antigenic peptides and polymers remain stable. They are also effective at enhancing the immunological response produced by intramuscular administration of DNA, making them a potential immunoadjuvant [3]. However, there have only been a few attempts to deliver oral vaccines using gold-based NPs and further research is needed to determine how they function as immune adjuvants after oral administration [84].

### 4.4. Niosomes

Niosomes are nonionic surfactant vesicles having a size range on a nanometric scale. Niosomes are prepared from a mixture of nonionic surfactants of the alkyl or dialkylpolyglycerol ether class and cholesterol followed by hydration in aqueous media. They serve as novel drug delivery carriers for drugs and are gaining importance in the oral delivery of vaccines [85]. The delivery of biomolecules has to face several obstacles while passing through the gastrointestinal tract. Such obstacles include changes in pH from the esophagus to the large intestine, proteolytic enzymes, and low epithelial permeability. The entrapment of antigens within niosomes helps to bypass the harsh gastric environment that causes the destruction of antigens. Moreover, the nanosize of niosomes also enhances the penetration of biomolecules through biological membranes. Target-oriented drug delivery can be acquired by attaching ligands over the surface of niosomes [86]. One study demonstrates the oral delivery of recombinant human insulin using a niosomal formulation. The encapsulation of insulin within the bilayer structure of polyoxyethylene alkyl ether-based niosomes protected it from niosomes proteolytic activity of α-chymotrypsin, trypsin, and pepsin in an in vitro study [87].

### 4.5. Vesicular Drug Delivery Systems

➢Exosomes

Exosomes, which are spherical vesicular structures derived from endosomes and range in size from 40 to 160 nm in diameter, are composed of several kinds of proteins and lipid bilayers. They are abundantly present in all physiological fluids and tissues and are released by various cell types [88]. Exosomes function as a tool for intercellular communication that transports materials from donor cells to recipient cells. Exosomes’ innate capacity to regulate intricate intracellular pathways has led to their usage in the management of several disorders. They can be altered to deliver different pharmacological and bioactive substances at specific sites [89]. Because they are isolated from the patient’s own cells, these vesicles have better biocompatibility, which makes them an important drug delivery vehicle. They can avoid the immune system’s early clearance because of their small size and ability to infiltrate deep tissues. Exosomes can be employed to carry medications across the blood–brain barrier in addition to improving drug qualities [90]. By activating a number of signaling pathways, mesenchymal stem cell (MSC)-derived exosomes are believed to play a remarkable role in the healing of wounds and many other inflammatory disorders [91].

Exosomes can be either naturally occurring or artificially produced. Animal- and plant-origin exosomes make up the second division of natural exosomes. Exosomes originating from animals are further split into normal exosomes and exosomes derived from tumors. All normal cell types, including T cells, beta cells, macrophages, dendritic cells, and mesenchymal stem cells, can release exosomes. For instance, mesenchymal stem cells can differentiate into multiple adult cell types. MSCs can readily adapt to the tumor microenvironment and have a high exosome production rate. Normal exosomes can be found in bile, breast milk, blood plasma, urine, and saliva [90]. The ability of exosomes generated from dendritic cells (DCs) to create antitumor antibodies is more impressive as compared to dendritic cells. Exosomes generated from DCs are thought to be a very effective platform for immunization in the near future since they contain major histocompatibility complex (MHC) class I and II in their membrane structure [92]. Chemotherapeutic drugs have been successfully delivered by exosomes derived from milk. Exosomes generated from different physiological fluids, however, can be used as both therapeutic and diagnostic agents. Tumor cells release large amounts of exosomes. The presence of a particular ligand on the surface of the exosomes indicates their origin from a certain type of donor cell. Exosomes from tumor cells play a crucial role in the development of tumors and the spread of disease. One of the methods for treating metastatic cancer is to prevent the generation of tumor exosomes. Plant-derived exosomes have also been investigated recently, along with animal-derived exosomes, and research has revealed that the two types of exosomes share structural similarities. Exosomes generated from ginger aid in the prevention of diseases of the liver. Exosomes isolated from grapes, carrots, and ginger also demonstrated anti-inflammatory properties and could keep the intestine’s physiological balance in the usual range [93]. Using exosomes, different proteins can be delivered to recipient cells, which can alter the biological behavior of the cells receiving the proteins. Such molecules can be loaded into exosomes using active and passive loading methods. Drugs can be loaded into exosomes via the active loading method known as electroporation, which uses an electric field to produce tiny holes in the membrane of the exosomes through which cargo can be introduced. In the passive loading technique, the drug can be loaded in a simple step of incubation. Exosome targeting action can be achieved via direct surface modification and manipulation in exosome donor cells. Manipulation in exosome donor cells is an endogenous loading method in which drugs are loaded in exosomes by the inherent mechanism of donor cells. The drug-loaded exosomes are then released from the donor cells by subjecting them to any mechanical or chemical stimuli. An exogenous loading technique is direct surface modification. In order to achieve target specificity, surface modifications such as chemical or genetic alterations are employed [94].

To deliver the COVID-19 vaccine, mRNA-based exosomes have recently been identified. Tsai et al. precisely fabricated the LSNME-SW1 vaccine, which generated wide immunities against COVID-19, and confirmed the utilization of exosomes for delivering the required mRNA into the host cell in vivo and in vitro [92].

➢Colloidosomes

The self-assembly of colloidal particles forms microcapsular structures known as colloidosomes. Such particles have a hold over the mechanical features of colloidosomes, which includes size, permeability, and strength [95]. Based on the kind of emulsion utilized in their formation, colloidosomes can be divided into various types. They can be o/w emulsion-based colloidosomes, w/o emulsion-based colloidosomes, and w/o/w emulsion-based colloidosomes [96]. Various methods have been reviewed for the formation of colloidosomes, and such methods include thermal annealing, gel trapping, covalent crosslinking, polyelectrolyte complexation, and polymerization of the droplet phase. Many drugs can be delivered by encapsulating them within the said carrier system, and site specificity can be achieved by attaching the ligand over them. Colloidosomes can encapsulate enzymes and vaccines within them and shield them from the unfavorable pH of the stomach when delivered through the oral route and provide a prolonged effect by sustained drug release [97].

➢Aquasomes

Aquasomes are self-organized nanoparticulate carrier systems composed of three basic layers. Among these layers, the first one is the innermost called the core, the second layer is a middle coating of carbohydrate material, and the third layer is of active pharmaceutical ingredient (API) moieties that are to be administered and are embedded in the carbohydrate layer [98]. The core is composed of ceramics that give uniformity and stability to the nanovesicular structure. The method used to synthesize the core depends upon the material selected. Above the core, there is a carbohydrate coating mostly of polyhydroxy oligomers to maintain humidity around the APIs moieties to prevent their degradation. Basically, these are prepared in three sequenced stages summarized as ceramic core formation, followed by carbohydrate wrapping of the core, and finally the adsorption of APIs as the outermost layer on carbohydrate [99]. Aquasomes offer regulated drug release and site-specific drug delivery. In addition to serving as oxygen carriers, they can be employed to deliver a variety of therapeutic agents such as pharmacological drugs, vaccines, proteins, DNA, and peptides. Approaches for administering vaccines based on aquasomes have various advantages. Adsorbed antigens on aquasomes have the capacity to elicit cellular and humoral immune responses [100]. According to research studies, different types of carbohydrates were used to coat the center of aquasomes [101]. In a new study, aquasomes were prepared by using a variety of carbohydrates for coating over the core, followed by the application of insulin to carbohydrates. Albino rats were used in an in vivo experiment, and it was observed that the rats’ blood sugar levels were significantly lowered. Different carbohydrate coatings had varying impacts on reduction; however, pyridoxal 5 phosphate demonstrated the greatest reduction when compared to trehalose. The drug’s gradual release from the carrier system was the cause of the prolonged drop in blood sugar level [102].

➢Polymersomes

Amphiphilic copolymer allows polymeric vesicles to self-assemble into polymersomes [103]. Polymers can be biodegradable and have high physicochemical stability when compared to surfactants. The development of polymer chemistry has made it simpler to manufacture a wide range of functional block copolymers, which has greatly sparked scientists’ curiosity in creating polymersomes as novel carrier systems [104]. Amphiphilic polymers, such as surface-active agents, can be combined to produce a variety of aggregates. Due to their excellent stability and adaptability, they can be immensely employed in therapeutic delivery. They can encapsulate hydrophilic or hydrophobic moiety individually or both at once to achieve the goal of combination therapy. Proteins in natural medicines and antigens can be shielded by polymersomes from harm in the GI lumen. For the oral delivery of insulin through polymerosomes, researchers prepared a copolymer of dextran-b-poly(lactide-co-glycolide). Insulin was tightly entrapped in polymersomes (>90%) and barely released in simulated gastric fluid (SGF), except for in simulated intestinal fluid (SIF). In diabetic rats, a significant hypoglycemic effect was achieved [105]. Polymerosomes have a significant impact in the field of nanomedicine, but still, they have not been utilized in the battle against COVID-19 [106].

➢Phytosomes

Aqueous-based phytomedicine–phospholipid complexes generate phytosomes, which are colloidal, small-sized vesicles, such as liposomes, having micellar or hexagonal nanostructures. They are sometimes referred to as pharmacosomes in the pharmaceutical industry [107,108]. Limitations associated with conventional drug delivery systems have been resolved by the discovery of advanced drug delivery vesicles. Nanometric size, amphiphilic nature, high entrapment efficiency, and high stability are the unique features of phytosomes that make them suitable carriers for the delivery of pharmaceuticals through the enteral route. Attachment of pharmaceuticals with phospholipid molecules, a principal component of phytosomes offers several benefits that include prevention of drug leakage from the carrier and slow and sustained drug release that enhance the efficacy of delivered medications [109].

➢Emulsomes

Emulsomes are a new type of vesicular system with an internal core consisting of solid fat and an outer phospholipid bilayer around it [110]. Cholesterol and soy lecithin are added to this structure to stabilize it. These are lipid-based delivery systems having the potential to deliver therapeutically active components with poor water solubility. They can be used to treat several complicated disorders, including fungal and viral infections, cancer, and dermal diseases that pose a serious risk to one’s health [111]. Emulsomes are great drug carriers for intravenous drug administration and can be used to give regulated medication delivery via the oral route by reducing GIT irritation [112,113]. Due to the fact that emulsomes may deliver sustained medication release for up to 24 h, they are superior to other vesicular systems. Emulsomes can deliver medicine targeted to the tumor location and reticuloendothelial system. They can be easily engulfed by the reticuloendothelial system (RES), which enables their use in the treatment of liver and spleen pathologies. By administering an anticancer medication encapsulated inside emulsomes, unwanted effects caused by chemotherapeutic agents can be reduced. This vesicular system can alter the drug’s pharmacokinetics and lengthen its elimination half-life. As a result, the dose needed to achieve the required response is lowered, in addition to lessening the number of unwanted side effects [114]. The incorporation of a vaccine within emulsomes maintains their antigenic integrity and stimulates their ability to provoke an immune response. Strong serum, lung, intestinal, and vaginal IgG and IgA responses were produced in mice after administering a mucosal vaccine through the intranasal route after enclosing a vaccine within emulsomes. The improved bioadhesiveness of emulsomes was seen after modifying the surface of the phospholipid bilayer of emulsomes by employing a mucoadhesive polymer [110].

➢Enzymosomes

Recently, enzymosomes have been reported as innovative vesicles. Enzymosomes are synthesized by incorporating enzymes within carriers similar to living cells having an enriched lipid background [115]. Enzymosomes receive particular attention in targeted drug delivery due to their ability to attach to specific receptor molecules. They are efficient enough to ameliorate the side effects of other conventional therapies due to their nonspecificity. Although enzymes can accelerate therapeutic effectiveness due to their high specificity for receptors, minimal permeation through lipid membranes tarnishes their effectiveness. Therefore, enzymosomes are synthesized in such a way that enzymes get a firm attachment with the lipid membrane to increase their stability, penetration, and effectiveness [116]. To combat pH fluctuations and to increase their effectiveness, enzymes are contained in or bound over lipid molecules that prevent their degradation and improve half-life. There are two ways to synthesize enzymosomes. In the first method, an enzyme is attached to the lipophilic functional group. In the second method, an enzyme is bound over the bilayer phospholipid. For brain-targeted delivery, biological membrane penetration is the most important factor. To encounter the blood–brain barrier enzymosomes can be the most important pharmaceutical carriers due to their higher lipophilicity like in the treatment of epilepsy because most medications cannot cross the lipid-rich barrier. Enzymosomes can increase the bioavailability of many drugs regardless of their solubility, molecular weight, and charge by increasing absorption through the gastrointestinal tract. Enzymosomes are designed in a way that can preserve the structural features and activity of enzymes and antigens/vaccines contained within them [117].

➢Sphingosomes

Complex stability issues associated with liposomes lay the foundation of sphingosomes’ formation. The instability of liposomes occurs due to the hydrolysis of the ester bond of phospholipids. This hydrolysis can be prevented using lipids having ether or amide linkage in place of an ester linkage. As a result, sphingolipids are currently employed to create stable a form of liposomes named sphingosomes [118] and are described as “concentric, dual-layered vesicular structure in which an aqueous region is fully encompassed by a lipid bilayer membrane mostly made of natural or synthetic sphingolipid” [119]. When compared to alternative formulations, liposomal formulations having a sphingomyelin-based lipid provide more benefits. Sphingosomes have better drug retention properties and are considerably more resilient to acid hydrolysis. Sphingosomes can be used as carriers for the delivery of various chemotherapeutic agents, vaccines, and other biological macromolecules. They can be supplied parenterally via intravenous, intramuscular, subcutaneous, and intra-arterial routes. In most situations, they can be supplied intravenously, though occasionally they can be inhaled. A large volume of sphingosome-based formulations can be administered into major central veins. Sphingosomes can also be administered orally or topically [120]. Various advantages and disadvantages associated with different nanocarriers are mentioned in Table 2.

## 5. Conclusions

The use of NP-based delivery systems makes it possible to administer vaccines orally, which is the preferred route due to its high patient compliance rate, capability for mass immunization, ease of use for less-trained staff, and lack of risk from needle-associated injuries. Prior to this, no vaccine has been delivered through the oral route, excluding live-attenuated or inactivated microorganism-based vaccines using nanoparticles. NPs enhance immunogenicity and provide stability and protection to antigens against the unsuitable gastric environment. The strong immune response of NPs could be by virtue of their nanosize that mimics the size of pathogens. Apart from size, other features, such as physicochemical characteristics, also influence the immune response of NPs. The stability of antigens in the GIT has been significantly improved by several mucoadhesive and stimuli-responsive biodegradable polymers, including PLGA, PGA, chitosan, and alginate alone or in combination. Liposomes and other vesicular delivery systems can incorporate both hydrophilic and hydrophobic components within them, and they can also be fabricated by covering them with different polymers to increase their stability against denaturation. Ligand-specific targeting to M cells, epithelial dendritic cells, and enterocytes can overcome poor permeation of antigens through the mucosal barrier of the intestine associated with oral delivery. It can be proposed that the hindrances accompanied by the GIT can be prevented by using nanotechnology that not only provides selective targeting to intestinal immune cells but also reduces toxic side effects.

## 6. Future Directions

Nanoparticle-based oral vaccines can be developed by selecting a suitable carrier into which vaccines can be incorporated that can provoke a better immune response. The current review suggests that vaccine delivery through the oral route has many more benefits as compared to the parenteral route. In addition to these advantages, the oral route for vaccine delivery also encounters certain challenges, which include the degradation of antigens in unfavorable conditions. Nanotechnology provides us with carriers that can overcome the limitations associated with the oral delivery of vaccines. The only remaining question is whether they can be made using economically viable methods to enable mass vaccination at a cheap cost, as oral immunization has demonstrated, which is especially useful in impoverished countries.

## Figures and Tables

**Figure 1 vaccines-11-00490-f001:**
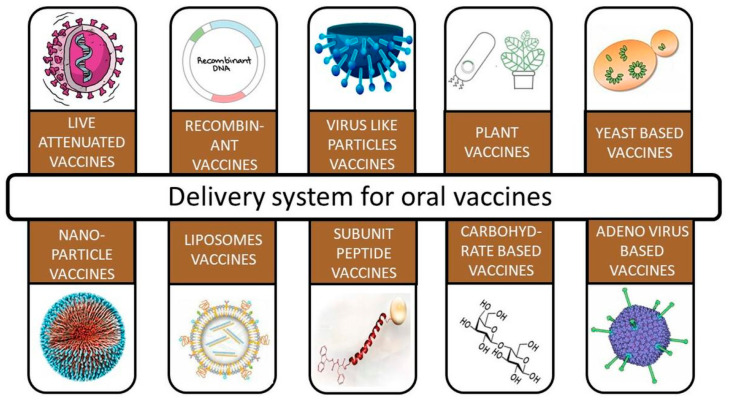
Various delivery systems for oral vaccines: Oral route for vaccines is not limited to a few options. It has engaged all the possible options and diversities in manufacturing a potent therapeutic product.

**Figure 3 vaccines-11-00490-f003:**
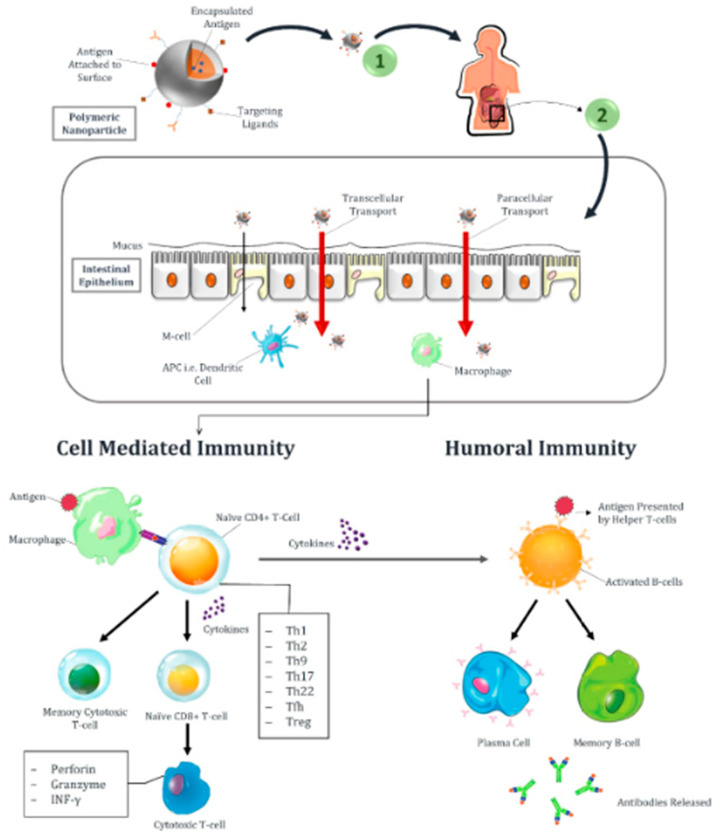
Transport mechanisms and enhanced activation of humoral and cell-mediated immune responses using nanoparticles.

**Table 1 vaccines-11-00490-t001:** List of oral vaccines available in market and under clinical trials.

Vaccines Available in Market [16]		
Disease	Oral Vaccines	Examples
Polio	Polio vaccine	Inactivated poliovirus vaccine, OPV
Gastroenteritis	Rota vaccine	Rotarix, Rotateq
Typhoid fever	Typhoid vaccine	Vivotif, Ty21a
Acute respiratory disease	Adenovirus vaccine	Adenovirus type 4 and 7
Cholera	Cholera vaccine	Vaxchora, Dukoral, Shanchol
Vaccines under Clinical Trial [17]		
COVID-19	COVID-19 vaccine	Under trial

**Table 2 vaccines-11-00490-t002:** Advantages and disadvantages of different carrier systems.

Nanoparticle System	Advantages	Limitations	References
Polymeric NPs	Sustained and controlled drug release, more stable than lipid-based NPs	Use of organic solvents, limited toxicity assessment in literature, difficulty in scale-up	[121,122]
Liposomes	Biocompatible, biodegradable, nontoxic, can enclose both hydrophilic and hydrophobic agents, can enhance bioavailability	Instability, leakage of enclosed antigen, rapid clearance through RES	[123]
Nanoemulsion	High loading capacity, greater stability, drug protection from degradation, economical	Low permeation and bioavailability, use of a large concentration of surfactant for stabilizing nanoparticles	[124,125]
Immunostimulating complexes	Provide site-specific delivery by attaching antibodies or in diagnostic immunological techniques	Inability to incorporate most soluble proteins as do not have exposed hydrophobic regions	[82]
Inorganic NPs	Biocompatibility	Instability and low loading capacity	[126]
Niosomes	Higher stability as compared to liposomes, high loading capacity, biocompatible, ability to entrap both hydrophilic and lipophilic agents, nonimmunogenic, enhance therapeutic effect	Lengthy preparation process, fusion hydrolysis, aggregation, leakage of entrapped drugs on poor storage conditions	[127]
Exosomes	Provide site-oriented delivery, minimal toxicity, maximum bioavailability, extend circulation time of components in blood	Less stable upon storage impurity	[128]
Colloidosomes	Easy fabrication, can encapsulate sufficient amount of antigen, mechanically robust, provide regulated release	Poor yield, sometimes show coalescence	[119]
Aquasomes	Enhance stability of biological entities, bypass rapid clearance through reticuloendothelial system	Preparation method is time-consuming	[98]
Polymersomes	More stable than liposomes, can entrap both hydrophilic and lipophilic components, better retain enclosed agent	Polymer can cause toxicity, difficulty in large-scale production	[129]
Phytosomes	Good drug retention in carrier, highly stable, deliver drug at the intended site, high loading capacity	Tend to fuse, aggregate, and hydrolyze upon storage	[130]
Emulsomes	Enhance bioavailability of agents having less aqueous solubility, protect antigen from denaturation in acidic pH, provide targeted delivery	Less stable	[130]
Enzymosomes	Provide targeted delivery, biodegradable, biocompatible, nontoxic, improved therapeutic effectiveness	Cost-ineffectiveness, phospholipids can be oxidized and fused and are liable to leak entrapped components from carrier system	[131]
Sphingosomes	Enhance efficacy of therapeutic moiety, less toxic, more stable, improves pharmacokinetic parameters	Low entrapment efficiency, expensive	[132,133]

## Data Availability

Not applicable.

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
