# Peer review of "Recent Developments in Oral Delivery of Vaccines Using Nanocarriers"

_vaccines, 2023, doi:10.3390/vaccines11020490_

Round 1

Reviewer 1 Report

Manuscript titled " Recent Developments in Oral Delivery of Vaccines using Nanocarriers" by Zafar et al, is a nice piece of writing. Author is able to discuss different types of nano particles available for drug or vaccine delivery. I think over all draft is in good shape and deserve place in journal. Before accepted for publication, I like that author should address the following minor issues.

1) Minor english editing is needed.

2) Author mentioned so many types of nano particles (just in text form). I think it is also important to include a figure  depicting all types of nano particles discussed in present review.

3) Author completely ignore one important and emerging plate form used or being tested for oral delivery of vaccines. Author should at least include a paragraph about yeast-based vaccines and include important references.

4) I think author also include yeast based system for oral vaccine delivery (even in figure 1)

5) In line 449, I think author mean PAIs

6) At few places authors directly used abbreviation without mentioning or describing it. Author should take care of this.

7) In section emulsomes author mentioned that, these particles can be used for fungal or viral infection. Is it specific for fungal and viral only??. Better to cite published study if available.

Author Response

Point 1: Minor English editing is needed.

Response 1: Language has been improved by a native English speaker with respect to language and grammar.

Point 2: Author mentioned so many types of nanoparticles (just in text form). I think it is also important to include a figure depicting all types of nanoparticles discussed in present review.

Response 2: We are thankful for valuable comment and a figure depicting all types of nanoparticles has been incorporated in revised manuscript.

Point 3: Author completely ignores one important and emerging platform used or being tested for oral delivery of vaccines. Author should at least include a paragraph about yeast-based vaccines and include important references.

Response 3: Discussion related to yeast-based vaccines has been incorporated in revised manuscript.

Point 4: I think author include yeast-based system for oral vaccine delivery (even in figure 1)

Response 4: Comment addressed and yeast-based system has been added in figure 1 in revised manuscript.

Point 5: In line 449, I think author mean PAIs

Response 5: Comment addressed. PAIs has been replaced with APIs in revised manuscript.

Point 6: At few places authors directly used abbreviation without mentioning or describing it. Author should take care of this.

Response 6: Abbreviations without their full form have been corrected in revised manuscript.

Point 7: In section emulsomes author mentioned that these particles can be used for fungal or viral infection. Is it specific for fungal and viral only? Better to cite published study if available.

Response 7: We are thankful for valuable comment, and it has been addressed by citing published studies.

Reviewer 2 Report

In this manuscript, the author suggests that vaccine delivery through oral route has much more benefits as compared to parenteral route. The topic is interesting. A minor revision is necessary for reconsideration to publish. The detailed comments on this manuscript are as follows:

1. What is the novelty of this work in comparison to the previous papers? Clarify the novelty from nanocarriers point of view in the manuscript.

2. Appropriate references must be cited in  the manuscript e.g. "due to the inherent challenges posed by the digestive system, only a tiny portion of presently approved vaccines are oral formulations." suitable references must be added to justify the results obtained.

3. English of the manuscript should be thoroughly checked and corrected.

Author Response

  1. What is the novelty of this work in comparison to the previous papers? Clarify the novelty from nanocarriers point of view in the manuscript.

Response 1: We are thankful for valuable comment, and it has been incorporated in revised manuscript. The current article has highlighted the latest updates regarding the oral vaccines and has compared various parameters, which can be beneficiary for later research.

  1. Appropriate references must be cited in the manuscript e.g., "due to the inherent challenges posed by the digestive system, only a tiny portion of presently approved vaccines are oral formulations." suitable references must be added to justify the results obtained.

Response 2: Comment has been addressed by citing specific references.

  1. English of the manuscript should be thoroughly checked and corrected.

Response 3: Language has been improved by a native English speaker with respect to language and grammar.

Reviewer 3 Report

The current review entitled "Recent developments in the oral delivery of vaccine using nanocarriers" describes the possible role of various types of nanoparticles as oral vaccine delivery systems. The manuscript is written and widely covers the topic. However, there are some important points that should be addressed during the revision of the manuscript. 

1. Adenoviral vectors have also been considered important delivery system for oral vaccines.  It should also be described with proper references.

2. A figure that shows the possible activation of humoral and cell-mediated antigen-specific immune responses should be presented.

3. Chitosan nanoparticles have recently been suggested as an effective carrier for oral vaccines. They should be discussed.

4.  Alpha-Galactosylceramide (alpha-GalCer) is a non-toxic oral adjuvant that can effectively induce T cell immunity. It should be mentioned with specific references.

Encapsulation of MERS antigen into α-GalCer-bearing-liposomes elicits stronger effector and memory immune responses in immunocompetent and leukopenic mice. J King Saud Univ Sci. 2022 Jul;34(5):102124.

Role of NKT Cells during Viral Infection and the Development of NKT Cell-Based Nanovaccines. Vaccines (Basel). 2021 Aug 26;9(9):949. 

Author Response

Point 1: Adenoviral vectors have also been considered important delivery system for oral vaccines. It should also be described with proper references.

Response 1: We are thankful for valuable comment, and it has been incorporated in revised manuscript.

Point 2: A figure that shows the possible activation of humoral and cell-mediated antigen specific immune responses should be presented.

Response 2: Comment addressed and its description is added in 1st paragraph of polymeric Nps.

Point 3: Chitosan nanoparticles have recently been suggested as an effective carrier for oral vaccines. They should be discussed.

Response 3: We are thankful for valuable comment and discussion related to chitosan nanoparticles has been incorporated in revised manuscript.

Point 4: Alpha-Galactosylceramide (alpha-GalCer) is a non-toxic oral adjuvant that can effectively induce T cell immunity. It should be mentioned with specific references.

Response 4: Comment has been addressed by citing specific references.

Round 2

Reviewer 3 Report

1. The authors have addressed some of my comments. Unfortunately, the authors did not address an important comment "Point 2: A figure that shows the possible activation of humoral and cell-mediated antigen specific immune responses should be presented." The response was addressed by a single sentence in the revised version of the manuscript. The authors must include a figure that shows the activation of antigen-specific humoral and T cell-mediated immune responses as the T cell response is critical for a successful anti-viral vaccine. 

2. The authors should also provide a table listing of approved and under trail oral vaccines. 

Author Response

Point 1: The authors have addressed some of my comments. Unfortunately, the authors did not address an important comment “Point 2: A figure that shows the possible activation of humoral and cell-mediated antigen specific immune responses should be presented.”The response was addressed by a single sentence in the revised version of the manuscript. The authors must include a figure that shows the activation of antigen-specific humoral and T cell- mediated immune responses as the T cell response is critical for a successful anti-viral vaccine.

Response 1: Comment addressed and a figure along with its description has been added in the revised manuscript.

Point 2: The authors should also provide a table listing of approved and under trial oral vaccines.

Response 2: We are thankful for valuable comment and a table listing of approved and under trial oral vaccines has been added in the revised manuscript.

Round 3

Reviewer 3 Report

The authors have responded to my comments in the revised manuscript.